# A Point-of-Care Faecal Test Combining Four Biomarkers Allows Avoidance of Normal Colonoscopies and Prioritizes Symptomatic Patients with a High Risk of Colorectal Cancer

**DOI:** 10.3390/cancers15030721

**Published:** 2023-01-24

**Authors:** Gonzalo Hijos-Mallada, Nuria Saura, Alberto Lué, Raúl Velamazan, Rocío Nieto, Mercedes Navarro, Samantha Arechavaleta, Eduardo Chueca, Fernando Gomollon, Angel Lanas, Carlos Sostres

**Affiliations:** 1Servicio de Aparato Digestivo, Hospital Clínico Universitario Lozano Blesa, 50009 Zaragoza, Spain; 2Instituto de Investigación Sanitaria (IIS) Aragón, 50009 Zaragoza, Spain; 3Departamento de Medicina, Universidad de Zaragoza, 50009 Zaragoza, Spain; 4CIBER Enfermedades Hepáticas y Digestivas (CIBERehd), 28029 Madrid, Spain

**Keywords:** colorectal cancer, advanced adenoma, inflammatory bowel disease, faecal haemoglobin, faecal calprotectin, faecal lactoferrin, faecal transferrin

## Abstract

**Simple Summary:**

Gastrointestinal symptoms are a common reason for endoscopic referral. Symptoms alone are unreliable predictors of colorectal cancer (CRC), but as CRC prognosis relies mainly in an early diagnosis, many of these patients undergo colonoscopy. Therefore, most colonoscopies performed in this setting are normal, exposing these patients to endoscopy-related risks and increasing the workload of endoscopic units. This work analyses a point-of-care (POC) qualitative faecal test that simultaneously detect four biomarkers, comparing it with a quantitative occult blood and calprotectin tests, in stool samples of symptomatic patients referred for colonoscopy. Our results indicate that the POC test may be a useful strategy to triage symptomatic patients, as patients with a negative result in the four biomarkers have a low probability of relevant pathology (94.8–100% for CRC). Moreover, a positive result in the four biomarkers was associated with high risk of serious pathology (70.6% were diagnosed with CRC or inflammatory bowel disease).

**Abstract:**

Most colonoscopies performed to evaluate gastrointestinal symptoms detect only non-relevant pathologies. We aimed to evaluate the diagnostic accuracy of a qualitative point-of-care (POC) test combining four biomarkers (haemoglobin, transferrin, calprotectin, and lactoferrin), a quantitative faecal immunochemical test (FIT) for haemoglobin, and a quantitative faecal calprotectin (FC) test in symptomatic patients prospectively recruited. Colorectal cancer (CRC), adenoma requiring surveillance, inflammatory bowel disease (IBD), microscopic colitis, and angiodysplasia were considered significant pathologies. A total of 571 patients were included. Significant pathology was diagnosed in 118 (20.7%), including 30 CRC cases (5.3%). The POC test yielded the highest negative predictive values: 94.8% for a significant pathology and 100% for CRC or IBD if the four markers turned negative (36.8% of the patients). Negative predictive values of FIT, FC, and its combination for diagnosis of a significant pathology were 88.4%, 87.6%, and 90.8%, respectively. Moreover, the positive predictive value using the POC test was 82.3% for significant pathology when all biomarkers tested positive (6% of the patients), with 70.6% of these patients diagnosed with CRC or IBD. The AUC of the POC test was 0.801 (95%CI 0.754-0.848) for the diagnosis of a significant pathology. Therefore, this POC faecal test allows the avoidance of unnecessary colonoscopies and prioritizes high risk symptomatic patients.

## 1. Introduction

Gastrointestinal symptoms are a common reason for consultation in both primary and secondary care. Most of these symptoms do not correlate well with the diagnosis of colorectal cancer (CRC) or other relevant pathologies [1,2]. However, due to the high incidence and mortality of CRC [3] and considering that its prognosis is largely conditioned by an early diagnosis [4], many of these patients undergo colonoscopy. Consequently, a significant percentage of colonoscopies performed in symptomatic patients are normal or detect only mild and benign pathologies [5], exposing these patients to the non-negligible risk of endoscopy-related complications [6] and increasing the waiting lists, in a situation in which the implementation of CRC screening programs [7] were already increasing the colonoscopy workload in most public health systems. Furthermore, the current COVID-19 pandemic has caused a substantial reduction in endoscopic activity, thus increasing the pressure on already overburdened endoscopic units [8]. Therefore, it is highly necessary to establish strategies to determine which endoscopic procedures should be prioritized or may be postponed. Regarding symptomatic patients, several strategies have been suggested and implemented. 

Current clinical practice guidelines recommend the use of the faecal occult blood test with a faecal immunochemical test (FIT) in symptomatic patients to triage referral for endoscopic examination, with a cut-off of 10 μg/g [9,10]. This statement is supported by several studies, reporting a high negative predictive value (NPV) for CRC and advanced adenoma [11,12]. FIT is a more accurate diagnostic tool that symptom-based referral criteria [13], and it may be used even in patients with high-risk symptoms [14,15]. It should be noted that some authors recommend a threshold of 20 μg/g, which is currently used in most CRC screening programs [16], since it appears to have similar diagnostic performance with less colonoscopies required [17]. After the onset of the COVID-19 pandemic, FIT has been widely recommended to triage symptomatic patients in need of colonoscopy, as it allows both a reduction in endoscopic requirements and to potentially minimize the long-term outcomes of diagnostic backlogs [18]. Faecal calprotectin (FC) is a widely used biomarker for diagnosis and monitoring of inflammatory bowel disease (IBD) [19]. The diagnostic yield of FC for the diagnosis of CRC and adenoma is less established, though it appears to have a high NVP, being more sensitive but less specific than FIT [20,21,22]. However, when combining FIT plus FC, the results of available studies are not consistent. While some studies have concluded that this combination does not provide a better diagnostic accuracy than FIT alone [23,24,25], others point to an enhanced diagnostic performance [26,27]. This disparity can be related with the tests used, as differences in FC levels are commonly found between assays from different manufacturers [28,29]. 

There is a paucity of studies regarding diagnostic performance of other biomarkers in symptomatic patients. Faecal transferrin has been suggested to increase the sensitivity of FIT alone in symptomatic patients for the detection of CRC and adenoma [30]. Conversely, in a CRC screening population, the combination of both markers did not improve the diagnostic performance of FIT [31]. Faecal lactoferrin correlates with inflammation similarly to FC [32]. Evidence regarding its diagnostic accuracy in CRC and adenoma detection is scarce, although one study reported similar results than FIT [33]. 

Quick point-of-care (POC) faecal tests have the main advantage of providing immediate results, so they can be extremely useful in primary care settings to select patients that require prompt endoscopic examination or secondary care referral. Proper training and quality assurance are mandatory to reduce inter-user variability [34,35].

In this study, we analysed the diagnostic accuracy of a quick POC qualitative faecal test, simultaneously detecting the four biomarkers previously mentioned, as well as a quantitative FIT and FC test, in a cohort of symptomatic patients referred for colonoscopy. The primary endpoint was to design new strategies enabling both the avoidance of unnecessary colonoscopies and to prioritize referrals in symptomatic patients. Secondary endpoints were to evaluate the diagnostic yield of each biomarker individually for the diagnosis of relevant colonic pathologies and for each specific diagnosis. 

## 2. Materials and Methods

We conducted a single centre, prospective observational study enrolling patients with gastrointestinal symptoms referred for diagnostic colonoscopy to the Endoscopic Unit of Hospital Clinico Universitario Lozano Blesa (Zaragoza, Spain). Ethical approval was granted by the local ethic committee (CEICA - Regional Ethical Committee of Aragón - study code PI21/182). Patients referred for colonoscopy due to gastrointestinal symptoms between March 2019 and July 2020, either from primary or secondary care, were consecutively enrolled. Each patient was contacted approximately 1 week before the colonoscopy was scheduled. Patients who agreed to participate were asked to collect a stool sample in a universal faecal container the day before starting the colonic preparation, keep it refrigerated, and bring it to the hospital on the day of the colonoscopy. Every patient signed a written informed consent before being included in the study. Patients who signed informed consent, brought the stool sample, and underwent colonoscopy were included in the study. Patients were excluded for the study if they were under 18 years old, colonoscopy was requested due to other indications (CRC screening, adenoma, or IBD follow-up; family history of CRC), or if the faecal sample returned was insufficient or unsuitable for analysis.

The following faecal tests were performed: FOB+Transferrin+Calprotectin+Lactoferrrin® (CerTest Biotec S.L, Zaragoza, Spain), a one-step chromatographic immunoassay for the simultaneous POC qualitative detection of human haemoglobin (hHb), human transferrin (hTf), human calprotectin (hCp), and human lactoferrin (hLf). Cut-off values of the test are 5.1 μg/g for hHb, 0.4 μg/g for hTf, 50 μg/g for hCp, and 10 μg/g for hLf.FIT, by FOB Turbilatex® (CerTest Biotec S.L, Zaragoza, Spain), a latex turbidimetric assay for the immunochemical quantitative detection of haemoglobin. A cut-off of 10 μg/g was chosen.FC, by Calprotectin Turbilatex® (CerTest Biotec S.L, Zaragoza, Spain), a latex turbidimetric assay, with a cut-off of 50 μg/g.

Both quantitative tests (FIT and FC) were analysed using the ChemWell-T® turbidimetry equipment (CerTest Biotec S.L, Zaragoza, Spain). The combined POC test was performed and read by trained investigators. No laboratory equipment is needed, only the specimen collection stick that fits in a collection tube and the test cartridge, enabling this test to be performed in an outpatient clinic or in primary care setting. Investigators performing these tests were blinded to the patient data and diagnosis. Colonoscopies were performed by experienced gastroenterologists. We defined significant colonic pathology as the presence of CRC, IBD, adenomas requiring endoscopic surveillance according to European Society of Gastrointestinal Endoscopy Guidelines [36] (≥5 adenomas, adenoma of size ≥10 mm or with high grade dysplasia, serrated lesions ≥10 mm or with dysplasia), microscopic colitis, and angiodysplasia. Non-significant findings included, aside from normal examinations, the following results: adenoma not requiring endoscopic surveillance and hyperplastic polyp. Uncomplicated diverticular disease and haemorrhoids were accounted as normal examinations. Gastroenterologists performing the endoscopic examinations were blinded to the results of the faecal tests.

A descriptive analysis of patients included was performed. Continuous variables were expressed as mean with standard deviation or median with interquartile range. The Kolmogorov–Smirnov test was used to assess if continuous variables followed a normal distribution. Qualitative variables were described with frequencies and percentages. Chi-square, Kruskal–Wallis, and Man–Whitney U tests were used to evaluate the relationship between different variables. Sensitivity, specificity, positive predictive value (PPV), NPV, and the area under receiver operator curve (AUC) were calculated for each faecal test (FIT, FC, and the four biomarkers that constitute the POC test), and for its possible combinations. The method of DeLong et al. was used to test the statistical significance of the differences between the AUCs [37]. A logistic regression analysis was performed to determine associations in terms of the odds ratio (OR) of each biomarker and its combinations adjusted by different demographic variables with the diagnosis of relevant pathologies. SPSS version 21® and MedCalc version 13.3® were used for statistical analysis.

## 3. Results

### 3.1. Study Population 

A total of 653 patients were contacted, of whom 608 agreed to participate in the study (93.1% participation rate). Thirty-seven patients were excluded due to exclusion criteria. Hence, 571 patients were included in the final analysis. 

The most common finding was normal colonoscopy (56.7%, 24/571). Adenomas not requiring surveillance were detected in 79 (13.8%) patients, and hyperplastic polyps in 50 (8.8%). Therefore, non-significant findings were reported in a 79.3% (453/571) of the colonoscopies performed. Significant colonic pathology was found in 118 patients (20.7%), that is, 30 CRC cases (5.3%), 15 IBD cases (2.6%), 53 patients diagnosed with adenoma requiring surveillance (9.3%), seven with microscopic colitis (1.2%), and 13 with colonic angiodysplasia (2.3%). The median age was 63 years (interquartile range 51.5–74.5 years), with the youngest patient being 18 and the eldest 90 years old. There was nearly an equal proportion of both sexes, although slightly more females were included (52.4%, 299/571). Most colonoscopies were requested by primary care (64.8%, 370/571). Previous history of rectal bleeding (28.9%, 165/571) was the commonest indication of referral, followed by chronic diarrhoea (20%, 114/571). A total of 27.7% (158/571) of the patients were under treatment with antiplatelets, anticoagulants, or nonsteroidal anti-inflammatory drugs (NSAIDs). The baseline characteristics of the participants and their association with the diagnosis of relevant colonic pathologies are summarized in Table 1.

### 3.2. Diagnostic Accuracy of Faecal Tests

Positivity rates were 21.5% (123/571) for FIT, 50.4% (288/571) for FC, and 56.2% (321/571) for the combination of both tests, considering this composite test positive if at least one of both tests turned positive (FIT or FC). Table 2 shows the diagnostic accuracy of quantitative FIT, FC, and the combination. Median values of FIT and FC results in each diagnosis are presented in Appendix A in the Appendix A.

Regarding the combined POC test, positivity rates were 23.8% (136/571) for hHb, 20.8% (119/571) for hTf, 58.3% (333/571) for hCp, and 13.3% (76/571) for hLf. In total, 36.8% (210/571) of the patients had a negative result in the four tests, whereas 6% (34/571) tested positive in the four biomarkers. Table 3, Table 4, and Figure 1 represent the diagnostic accuracy of the POC faecal test when analysing each biomarker individually and in combination. 

Diagnostic accuracy of FIT, FC, and the combined POC test for the diagnosis of a significant colonic pathology and CRC, stratified by the main presenting symptom, is presented in Appendix A in the Appendix A. 

Patients under any of the concomitant treatments analysed (NSAIDs, antiplatelets, or anticoagulants) had a higher positivity rate of FIT (28.5% vs 18.9%, *p* = 0.01, only significant in anticoagulants users, 43.3% vs 20.3%, *p* = 0.005), FC (65.8% vs 44.5%, *p* < 0.001, only reaching significance in antiplatelets users, 68.7% vs 47.2%, *p* < 0.001), and the POC test with ≥1 tests (79.7% vs 56.9%, *p* < 0.01, 76% vs 61.9% in NSAIDs users, *p* = 0.033, 80.2% vs 60.2% in antiplatelets users, *p* < 0.001, 90% vs 61.7% in anticoagulants users, *p* = 0.001), ≥2 tests (44.9% vs 26.6%, *p* < 0.01, 45.3% vs 29.3% in antiplatelets users, *p* = 0.003, 60% vs 30.1% in anticoagulants users, *p* = 0.01), and ≥3 tests cut-offs (20.3% vs 13.5%, *p* = 0.034, not reaching significance in any of the specific drugs). The positivity rate of the POC test when four positive tests was used as a cut-off was higher in patients taking concomitant treatments, but this difference did not reach significance (7.6% vs 5.3%, *p* = 0.202). Nevertheless, no significant differences were found in the false positive rate of FIT (42.2% vs 46.1%, *p* = 0.408), FC (71.1% vs 71.2%, *p* = 0.549), or the POC test with ≥1 tests (69.8% vs 70.6%, *p* = 0.483), ≥2 tests (59.1% vs 50%, *p* = 0.146), and ≥3 tests cut-offs (50% vs 35.7%, *p* = 0.139) in patients taken any of these drugs. The false positive rate of the POC test with a cut-off of four positive tests was significantly higher in patients under concomitant treatments (41.6% vs 4.7%, *p* = 0.014). Only 6/34 patients with four positive tests were not diagnosed with a significant pathology, and five of them were under concomitant treatments (three were taking antiplatelets, one NSAIDs, and one anticoagulants). Consequently, considering only patients not taking any concomitant treatment, the PPV of a positive result in the four biomarkers of the POC test increases up to 95.4% (21/22). 

### 3.3. Receiver Operator Curves Analysis

The AUCs for diagnosis of significant colonic pathology were 0.797 (95%CI 0.747–0.846) for FIT, 0.651 (95%CI 0.595–0.708) for FC, and 0.690 (0.637–0.743) for the combination of both tests. The AUC of FIT was significantly higher than both the AUC of FC and the AUC of the combination of both tests (*p* < 0.01). The AUCs of each biomarker of the combined POC test and of the combination of the four biomarkers are shown in Table 5. AUCs of all the biomarkers explored in this study for the diagnosis of CRC, adenoma requiring surveillance, and IBD are summarized in Appendix A, available in the Appendix A. 

The AUC of the combination of the four tests was calculated considering the four possible cut-offs according to the number of positive tests. The AUC of the combined POC test was significantly higher than the AUC of FC, hTf, hCp, and hLf, although no significant difference was found with the AUC of hHb (*p* = 0.076) and with quantitative FIT (*p* = 0.848). The AUCs of the POC test compared with quantitative FIT and FC are represented in Figure 2. 

## 4. Discussion

This study evaluated the diagnostic accuracy of three different faecal tests (FIT, FC, and a POC test combining four biomarkers) in symptomatic patients referred for colonoscopy for either primary or secondary care. The first result that should be highlighted is that only 20.7% of the patients included were diagnosed with a relevant colonic pathology, being more than a half of the colonoscopies performed completely normal (56.7%). These figures are similar to those reported by other studies [13,14,17,23,24,27], hence strategies to accurately triage symptomatic patients are urgently needed to avoid these unnecessary colonoscopies and ease pressures on endoscopic units.

Quantitative FIT, with a cut-off of 10 μg/g, showed a high NPV for CRC (98.7%) and for significant colonic pathology diagnosis (88.4%), with the highest AUC compared with the other test analysed, and similar (no statistically significant differences) to the AUC of the combined POC faecal test comprising four biomarkers. These findings are consistent with prior studies [14,17], confirming that FIT is a helpful test to triage symptomatic patients. Combining FC with FIT did not improve the diagnostic accuracy of FIT alone, a conclusion also reached by similar studies [23,24,25]. Nevertheless, it should be mentioned that FC provided better sensitivity for CRC (83.3% vs 80%) and adenoma requiring surveillance (60.4% vs 47.2%) detection than FIT. This finding is not consistent with the results of previous comparative studies, with most suggesting than FIT is a more sensitive biomarker than FC for CRC diagnosis [23,27]. However, evidence regarding FC diagnostic yield for CRC and adenomas is not as solid as in FIT, and sensitivity values vary widely between studies due to high heterogeneity, with some authors suggesting that FC may be a more sensitive but less specific biomarker than FIT for CRC diagnosis [20]. In any case, both in these mentioned studies [23,24,25] and in our population, a non-negligible number of patients with a significant pathology had negative FIT results. In our cohort, 50 patients presented with a significant pathology and were FIT negative (six CRC, four IBD, 28 adenomas requiring surveillance, seven microscopic colitis, and five angiodysplasia). Even using any detectable FIT as cut-off, as has been suggested by prior studies in symptomatic patients [24], 15 patients with significant pathology would have been undetected, including four cases of CRC. Similarly, a study performed in symptomatic patients referred for colonoscopy reported a CRC miss rate of 14% using FIT with a cut-off of 10 μg/g [38]. 

Therefore, strategies to improve FIT diagnostic accuracy in this setting are necessary to avoid these underdiagnosed cases without implying a high rate of false positives. In this direction, the POC test combining four biomarkers may be a useful alternative as it has several potential benefits, even though its overall diagnostic accuracy (AUC) is not better than the quantitative FIT. As it is a POC test, it is easy to perform and interpret, does not need any laboratory equipment, and gives immediate results. This could be advantageous especially in outpatient clinics and primary care settings, reducing the interval between initial consultation and endoscopic examination, and thus avoiding a delayed diagnosis of CRC cases, which could potentially worsen its prognosis [39]. Indeed, this test can be performed even by patients themselves, communicating the result to the medical team. This application of the POC test would minimize the need of hospital visits, which has been advocated as a strategy to minimize transmission during the COVID-19 pandemic [8]. Besides this advantage, this test may also be integrated in telemedicine strategies, potentially alleviating the pressure on healthcare systems, and promoting patient’s self-management [40]. This test, as it combines markers with high sensitivity and others with high specificity, allows the detection of both patients with low and high risks of colonic pathology. As it has been previously reported [20], hCp had the highest sensitivity value for significant pathology (83.1%), but with a low specificity (43.6%). In contrast, hLf (92.3%) and hHb (87.4%) yielded the highest specificity. hHb, in parallel with the results of the quantitative test, was the test with the best diagnostic accuracy, with an AUC significantly higher that any of the other three tests and with no significant differences with the AUC of the combined POC test. However, likewise in the quantitative test, hHb alone underdiagnosed 39 cases of significant pathology including four cases of CRC. This is a lower number of undetected cases compared with quantitative FIT, which is related to the fact that this quantitative test has a threshold of 5.1 μg/g, which is lower than the cut-off chosen for the quantitative FIT (10 μg/g).

The main advantage of the test comes when combining the results of the four biomarkers. Having a negative result in the four biomarkers showed the highest NPV for significant colonic pathology (94.8%), being as high as 100% for both CRC and IBD. Only 11 cases of significant colonic pathology were underdiagnosed, being nine adenomas requiring surveillance and two cases of microscopic colitis. The NPV of a negative result in the four biomarkers for the diagnosis of significant pathology was higher in females (97.3% vs 91.9% in males) and in younger patients (98.5% in patients under 50 years old vs 93% in older patients). This better performance of the faecal test in women and younger patients has been previously reported in similar studies [41]. A triage strategy based in the POC test would have avoided 36.8% (210/571) of colonoscopies without missing any CRC or IBD case. One might believe that this is not a real advantage as several strategies have been published previously to avoid unnecessary colonoscopies in symptomatic patients, reporting NPVs comparable to the figure reached by our test. As it has been mentioned, FIT alone [12,14,15], FIT combined with FC [26,27], and FIT combined with symptoms-based prediction models [13] have proved to be useful in this setting. Nevertheless, this study provides a new strategy, as this POC faecal test allows not only the avoidance of unnecessary colonoscopies, but also to select patients at a very high risk of significant pathology in whom an urgent colonoscopy would be warranted. In the 34 patients with a positive result in the four biomarkers, the PPV for significant colonic pathology was 82.3%. CRC was diagnosed in 15 of these patients and IBD in nine (70.6% PPV for CRC or IBD). According to this data, 6% (34/571) of the total colonoscopies should have been prioritized and would have diagnosed the 50% of all the CRC cases and 60% of IBD cases in our cohort. 

Intermediate results of the combined test (2–3 tests positive) turned into progressive lower sensitivity and higher specificity and vice versa, as is shown in Table 4 and Figure 1, so the cut-off can be chosen according to the availability of endoscopic resources and the clinical suspicion. This is a non-invasive, inexpensive, and easy to perform test, so in case of persistence of symptoms it can be repeated. Regardless of the cut-off chosen, we can state that patients with no positive tests have low risk of significant pathology and colonoscopy can be avoided or postponed, and patients with four positive tests should undergo urgent/preferent colonoscopy. This strategy would have improved the diagnostic approach in 42.8% of the patients in our cohort.

Regarding the limitations of this study, even though we have included more than 500 patients in a single-centre population, the low prevalence of relevant pathologies in symptomatic patients has led to a low number of CRC or IBD cases. Therefore, specific analysis of the diagnostic yield of the studied tests for these individual diagnoses should be interpreted with caution. Our results may lead to multicentric large scale studies, providing a higher sample size. We have not performed an economic evaluation of the implementation of these tests to triage symptomatic patients, which is also a major limitation of this study. Previous studies conducted in similar populations suggested that combining faecal biomarkers is a more cost-effective strategy than using a single faecal test, and especially more than direct evaluation by colonoscopy [27]. Furthermore, faecal tests are usually better accepted by symptomatic patients compared with endoscopic examination [42], thus strategies based on these tests potentially could be applied more universally. Regarding the influence that concomitant treatments (NSAIDs, antiplatelets, and anticoagulants) may have had in these results, we have found a higher FIT positivity rate in anticoagulant users without differences in the false positive rate, as has been reported previously [43]. FC, as expected [29], and the POC test using 1–3 tests as cut-off yielded higher positivity rates in patients under concomitant treatments, again without significant impact in its diagnostic yield. Nevertheless, it should be noted that most patients with false positive results of the POC test with four positive tests were under any of these concomitant treatments (5/6). This finding suggests that having a positive result in the four biomarkers of the POC test implies a significantly higher risk of pathology (>90%) in patients not taking NSAIDs, antiplatelets, or anticoagulants. These results should be interpreted with caution as this study was not designed to assess the impact of drugs on biomarkers diagnostic yield, and the number of patients with four positive tests is low.

Another potential limitation is that it might be believed that FIT has no indication in patients consulting with rectal bleeding, which was the most frequent symptom in our cohort (28.9%, 165/571). However, FIT was negative in 75.8% of these patients (125/165) and the four biomarkers of the POC test were negative in 37% (61/165), with percentages almost identical to the figures obtained in the whole cohort (78.5% and 36.8%). FIT reached a sensitivity of 100% for CRC diagnosis (69.4% for significant pathology) in patients consulting with rectal bleeding, and the NPV of the POC test for diagnosis of significant pathology was 95.1% (Appendix A). A similar conclusion was reached by a multicentric study including more than 3000 patients with rectal bleeding (sensitivity of FIT for CRC diagnosis 96.6% vs 86.3% in patients consulting with other symptoms) [44]. Therefore, we can conclude that either FIT or other faecal biomarkers should be used to rule out CRC in patients with rectal bleeding.

## 5. Conclusions

In conclusion, the need for accurate strategies to triage symptomatic patients referred for colonoscopy should be again highlighted, as the percentage of normal examinations in this population is currently excessively high. 

Our study supports the utility of FIT as a diagnostic tool in symptomatic patients, given that it has a high NPV for CRC and significant pathologies. However, using only FIT entails a relevant number of undetected cases. We provide new evidence about the diagnostic accuracy of a POC faecal test combining four biomarkers that, although has not yielded a better overall diagnostic accuracy than quantitative FIT, may allow both the avoidance of unnecessary colonoscopies in low-risk patients, minimizing undetected cases, and the prioritization of high risk symptomatic patients. Further research is needed to explore the health economic benefits of using this test.

## Figures and Tables

**Figure 1 cancers-15-00721-f001:**
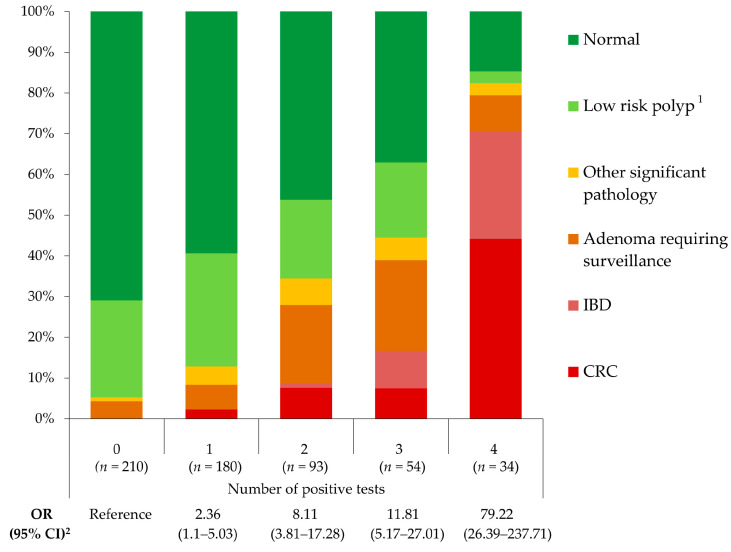
Colonoscopy findings regarding the number of positive results of the four-biomarker combined POC faecal test. ^1^ Low risk polyp includes adenomas with no indication of follow-up and hyperplastic polyps. ^2^ OR (95%CI): Odds Ratio (95% confidence interval), risk of presenting significant colonic pathology for each result of the test compared to having the four biomarkers negative, adjusted by age and sex.

**Figure 2 cancers-15-00721-f002:**
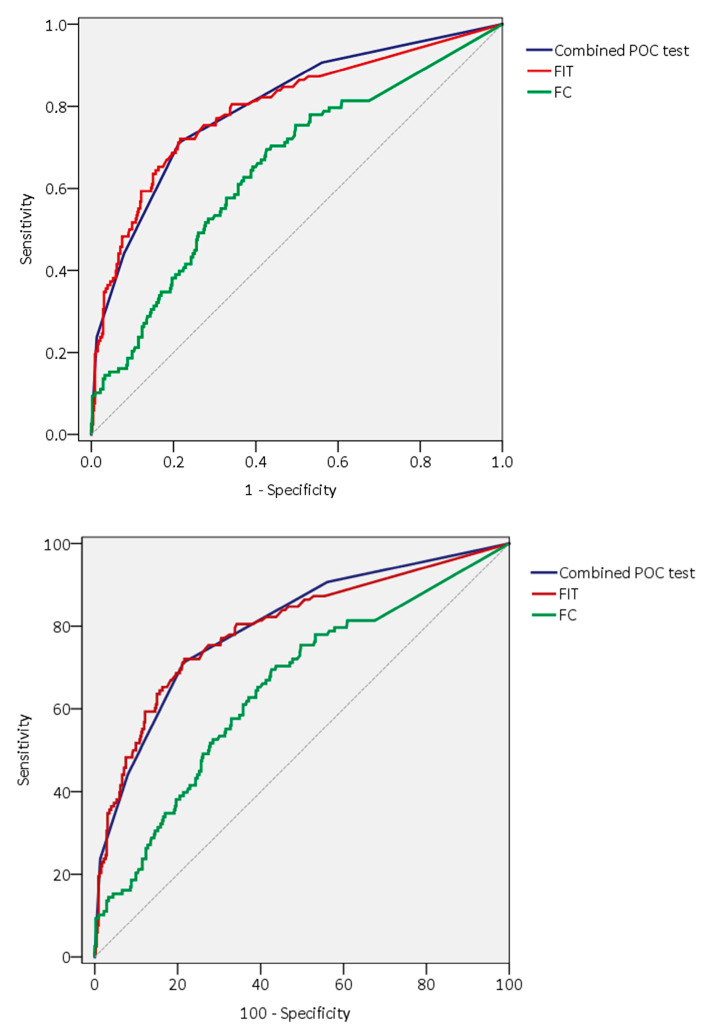
Receiver operation curves of FIT, FC, and the four-biomarker combined POC faecal test for the diagnosis of significant colonic pathologies.

**Table 1 cancers-15-00721-t001:** Baseline demographic data and their association with significant pathologies.

Demographic Data		Significant Pathology *n* = 118	Non-Significant Findings *n* = 453	*p*-Value Univariant	*p*-Value Multivariant OR (CI 95%) ^2^
Median age in years (interquartile range)		70 (59.5–80.5)	60 (48.5–71.5)	*p* < 0.01	*p* < 0.01 1.04 (1.02–1.05)
Gender	Male Female	67 (24.6%) 51 (17.1%)	205 (75.4%) 248 (82.9%)	*p* = 0.017	*p* = 0.039 1.57 (1.03–2.42)
Concomitant treatments	Any of the following NSAIDs ^1^ Acetylsalicylic acid Other antiplatelets Vitamin K antagonist Direct oral anticoagulants	42 (26.6%) 9 (18%) 21 (27.3%) 4 (22.2%) 7 (43.8%) 6 (42.9%)	116 (73.4%) 41 (82%) 56 (72.7%) 14 (77.8%) 9 (56.3%) 8 (57.1%)	*p* = 0.02	*p* = 0.984
Department requesting colonoscopy	Primary Care Gastroenterology General Surgery Other	89 (24.1%) 18 (12.9%) 4 (10.8%) 7 (29.2%)	281 (75.9%) 122 (87.1%) 33 (89.2%) 17 (70.8%)	*p* = 0.019	*p* = 0.229
Indication	Rectal bleeding Chronic diarrhoea Abdominal pain Change in bowel habits Anaemia / Iron deficiency Other	36 (21.8%) 18 (15.8%) 17 (17.5%) 16 (17.6%) 29 (32.6%) 2 (20%)	129 (78.2%) 96 (84.2%) 80 (82.5%) 75 (82.4%) 60 (67.4%) 13 (80%)	*p* = 0.064	-

^1^ NSAIDs: nonsteroidal anti-inflammatory drugs. ^2^ OR (95%CI): Odds ratio (95% confidence interval), risk of presenting with a significant colonic pathology in each category.

**Table 2 cancers-15-00721-t002:** Diagnostic accuracy of FIT, FC, and the combination.

Diagnosis	Test	True Positives	False Negatives	Sensitivity	Specificity	PPV	NPV	OR (95%CI) ^1^
Significant Pathology (*n* = 118)	FIT FC FIT or FC	68 83 95	50 35 23	57.6% 70.3% 80.5%	87.9% 54.8% 50.1%	55.3% 28.8% 29.6%	88.4% 87.6% 90.8%	8.7 (5.4–13.9) 2.3 (1.5–3.7) 3.5 (2.1–5.8)
Colorectal Cancer (*n* = 30)	FIT FC FIT or FC	24 25 28	6 5 2	80% 83.3% 93.3%	81.7% 51.4% 45.9%%	19.5% 8.7% 8.7%%	98.7% 98.2% 99.2%	15.3 (6.1–38.9) 4.2 (1.6–11.3) 9.6 (2.3–41.2)
Inflammatory Bowel Disease (*n* = 15)	FIT FC FIT or FC	11 13 14	4 2 1	73.3% 86.7% 93.3%	79.9% 50.5% 44.8%	8.9% 4.5% 4.4%	99.1% 99.3% 99.6%	16.3 (4.8–55.9) 9 (2–41.4) 15.3 (2–119.4)
Adenoma Requiring Surveillance (*n* = 53)	FIT FC FIT or FC	25 32 39	28 21 14	47.2% 60.4% 73.6%	81.1% 50.6% 45.6%	20.3% 11.1% 12.1%	93.8% 92.6% 94.4%	3 (1.7–5.6) 1.2 (0.7–2.1) 1.8 (1.1–3.2)

^1^ OR (95%CI): Odds ratio (95% confidence interval), risk of presenting the outcome with a positive result of the test compared to a negative result, adjusted by age and sex.

**Table 3 cancers-15-00721-t003:** Diagnostic accuracy of each biomarker of the combined POC faecal test.

Diagnosis	Test	True Positives	False Negatives	Sensitivity	Specificity	PPV	NPV	OR (95%CI) ^1^
Significant Pathology (*n* = 118)	hHb hTf hCp HLf	79 53 98 41	39 65 20 77	66.9% 44.9% 83.1% 34.8%	87.4% 85.4% 48.1% 92.3%	58.1% 44.5% 29.5% 53.9%	91% 85.6% 91.6% 84.4%	12.6 (7.8–20.3) 5.3 (3.3–8.6) 3.7 (2.2–6.3) 5.7 (3.3–9.7)
Colorectal Cancer (*n* = 30)	hHb hTf hCp HLf	26 18 28 18	4 12 2 12	86.7% 60% 93.3% 60%	79.7% 81.3% 43.6% 89.3%	19.1% 15.1% 8.4% 23.7%	99.1% 97.4% 99.2% 97.6%	22.2 (7.5–65.7) 6.7 (3.1–14.6) 8.4 (1.9–36.4) 10.8 (4.9–24)
Inflammatory Bowel Disease (*n* = 15)	hHb hTf hCp HLf	14 10 15 14	1 5 0 1	93.3% 66.7% 100% 93.3%	78.1% 80.4% 42.8% 90.5%	10.3% 8.4% 4.5% 21.2%	99.8% 98.9% 100% 99.8%	75.3 (9.4–600.6) 8.1 (2.7–24.3) - 142.6 (18–1128.2)
Adenoma Requiring Surveillance (*n* = 53)	hHb hTf hCp HLf	31 20 38 6	22 33 15 47	58.5% 37.7% 71.7% 11.3%	79.7% 80.9% 43% 86.5%	22.8% 16.8% 11.4% 7.9%	94.9% 92.7% 93.7% 90.5%	4.4 (2.4–8.1) 2.7 (1.4–5) 1.4 (0.7–2.7) 0.5 (0.2–1.4)

^1^ OR (95%CI): Odds ratio (95% confidence interval), risk of presenting the outcome with a positive result of the test compared to a negative result, adjusted by age and sex.

**Table 4 cancers-15-00721-t004:** Diagnostic accuracy of the combined POC faecal test, considering the four possible cut-offs according to the number of positive tests.

Diagnosis	Positive Tests	True Positives	False Negatives	Sensitivity	Specificity	PPV	NPV	OR (95%CI) ^1^
Significant Pathology (*n* = 118)	≥1 test ≥2 tests ≥3 tests 4 tests	107 84 52 28	11 34 66 90	90.7% 71.2% 44.1% 23.7%	43.9% 78.6% 92% 98.7%	29.6% 46.4% 59.1% 82.3%	94.8% 91.3% 86.3% 83.2%	6.5 (3.4–12.6) 8.1 (5.1–12.9) 8.3 (5–13.9) 25 (9.5–65.7)
Colorectal Cancer (*n* = 30)	≥1 test ≥2 tests ≥3 tests 4 tests	30 26 19 15	0 4 11 15	100% 86.7% 63.3% 50%	38.8% 71.4% 87.3% 96.5%	8.3% 14.4% 21.6% 44.1%	100% 99% 97.7% 97.2%	- 13.9 (4.7–40.9) 10.2 (4.6–22.7) 24.9 (10.4–59.5)
Inflammatory Bowel Disease (*n* = 15)	≥1 test ≥2 tests ≥3 tests 4 tests	15 15 14 9	0 0 1 6	100% 100% 93.3% 60%	37.8% 70.1% 86.7% 95.5%	4.2% 8.3% 15.9% 26.5%	100% 100% 99.8% 98.9%	- - 126.5 (15.9–1008) 41.5 (12.6–136.9)
Adenoma Requiring Surveillance (*n* = 53)	≥1 test ≥2 tests ≥3 tests 4 tests	44 33 15 3	9 20 38 50	83% 62.3% 28.3% 5.7%	38.8% 71.4% 85.9% 94%	12.2% 18.2% 17.1% 8.8%	95.7% 94.9% 92.1% 90.7%	2.5 (1.2–5.3) 3.3 (1.8–6.1) 1.8 (0.9–3.6) 0.7 (0.2–2.5)

^1^ OR (95%CI): Odds ratio (95% confidence interval), risk of presenting the outcome with a positive result of the test compared to a negative result, adjusted by age and sex.

**Table 5 cancers-15-00721-t005:** AUCs (95%CI) of each biomarker of the combined POC faecal test and the combination for the diagnosis of significant colonic pathologies.

Diagnosis	hHb	hTf	hCp	hLf	Combination of Four Tests
Significant pathology	0.772 (0.718–0.825)	0.652 (0.592–0.712)	0.656 (0.604–0.707)	0.635 (0.573–0.696)	0.801 (0.754–0.848)
*p* value ^1^	*p* = 0.076	*p* < 0.01	*p* < 0.01	*p* < 0.01	Reference
*p* value ^1^	Reference	*p* < 0.01	*p* < 0.01	*p* < 0.01	*p* = 0.076

^1^ *p* value refers to the comparison of the AUC of each biomarker against the reference.

## Data Availability

The data that support the findings of this study are available from the corresponding author upon reasonable request.

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
