# Peer review of "A Point-of-Care Faecal Test Combining Four Biomarkers Allows Avoidance of Normal Colonoscopies and Prioritizes Symptomatic Patients with a High Risk of Colorectal Cancer"

_cancers, 2023, doi:10.3390/cancers15030721_

Round 1
Reviewer 1 Report
The study by Gonzalo et al. is a very interesting and relevant study that aims to analyse Point of care qualitative faecal tests (Hemoglobin, transferrin, Claprotectin and Lactoferrin) and compare it with FIT and FOC (standard of care tests) in patients refreed for colonoscopy. The study has a specific objective of identifying biomarkers that could help prevent unneccessary colonoscopy in patients. This is an excellent thought process as colonoscopies are not only highly invasive but are expensive and extermely uncomfortable for patients. A proper preparation and counselling is required to assist the patient in the process. Thus finding non-invasive markers could be the key to patients care and management.
The authors have defined all methods in detail with results presented in an intelligeble manner. The tables and figures are easy to understand and provide comprehensive details required for the reader to understand the significance of each biomarkers from a sensitivity, specificity NPV and PPV perspective. AUC is well presented to support the results.
The conclusion supported the results with AUC for the combined POC significantly highere than the AUC for FCM hTf and hCp.
The study requires two clarifications:
Firstly, is it possible that the results of the POC tests affected by the use of anti-platelets, anticoagulants and NSAIDs? I see that the uni and multivaraite analysis does not show any effect but was any seperate analysis with Man whitney or Chi square show otherwise. This is an important aspect and needs clarification in the manusript.
Secondly, the results from the table clearly shows >2 tests have high sensitivity and specificity for significant pathology, ARS and CRC while > 3 tests have high specificity and sensitivity for IBD. This is important because if 2 or 3 tests are good enough to identify these pathologies , then why would are four tests neccessary? I see that the AUC for 4 tests is high but did the authors run AUC for a combination of 2/ 3 tests in comparison to the 4 tests to see if a lower number of tests can be helpful as well. Pleaseclarify if this was done.
The study deserves high ranking due to its significance for patients care.
antiplatelets, anticoagulants or nonsteroidal anti-inflammatory drugs (27.7%)- what could be the role of these drugs on the POC tests?
antiplatelets, anticoagulants or nonsteroidal anti-inflammatory drugs (27.7%)- what could be the role of these drugs on the POC tests?
Reviewer 2 Report
This is an interesting paper and point of care testing has important benefits.
Comments for improvement:
1. Within the methods section the description of the test(s) should be made clearer. It was a little confusing as what was laid out in the text and in the Tables e.g. in Table 2 didn't concur with text in methods section.
please make clear the specific test and combination
Further please outline what is meant by FOB - is this guaic based test for haemoglobin ?
2. Looking at Figure 2 of the AUC plot, the combination test appears to perform just as well as FIT alone; In the discussion the authors seem to prefer the POC and am uncertain of rationale based on the AUC figure? Please provide some detail here.
3. In Table 2, FC appears to have better sensitivity than FIT alone for CRC and adenomas - this would differ from other published series. What were the cut offs applied here?
More clarity and detail around this needs to be made in the Discussion
4. Would be helpful to outline potential costs for POC and FIT as well as difference in uptake (%) by patients for faecal testing - this would determine if such testing can be applied more universally.
5. Please add in the results section test accuracy against symptoms (rectal bleeding) - this was briefly discussed but not in detail
Also in the group with iron deficiency anaemia - this was a bigger cohort (32%)
Author Response
Response to Reviewer 2 Comments
This is an interesting paper and point of care testing has important benefits. Comments for improvement:
We would like to thank the reviewer for this comprehensive evaluation and the efforts made towards improving our manuscript. We largely agree with the suggestions made and considered them in a revised version of the manuscript.
Point 1. Within the methods section the description of the test(s) should be made clearer. It was a little confusing as what was laid out in the text and in the Tables e.g. in Table 2 didn't concur with text in methods section. Please make clear the specific test and combination Further please outline what is meant by conFOB - is this guaic based test for haemoglobin?
As the reviewer recommends, we have performed minor modifications in the Methods and Results sections in order to make these points clearer. Three faecal tests have been analyzed in our study: a quantitative faecal immunochemical test for hemoglobin (FIT), a quantitative faecal calprotectin test (FC) and the POC test simultaneously detecting four biomarkers (referred in the whole manuscript as hHb, hTf, hCp and hLf). Results are presented separating those of the quantitative tests (Table 2) from the POC test (Table 3, 4 and Figure 1).
“FOB” is only written in our manuscript when mentioning “FOB Turbilatex®” and“FOB+Transferrin+Calprotectin+Lactoferrrin®”, which are the commercial names of the tests used. Both tests detect hemoglobin using immunochemical methods, and therefore, throughout the whole article the quantitative fecal occult blood test used is mentioned as “FIT”. As the reviewer suggests, we have clarified in Methods that the quantitative occult blood test used is performed with an immunochemical method to avoid misinterpretations.
Point 2. Looking at Figure 2 of the AUC plot, the combination test appears to perform just as well as FIT alone; In the discussion the authors seem to prefer the POC and am uncertain of rationale based on the AUC figure? Please provide some detail here.
As the reviewer correctly highlights, the manuscript and its most visible sections (title, abstract, conclusions) are focused on the diagnostic yield of the POC test, which indeed has not shown a better diagnostic accuracy than FIT alone (no significant differences in AUCs). FIT is a validate surrogate marker with compelling evidence in symptomatic patients, providing a high sensitivity and AUC for CRC and other relevant pathology diagnosis. The results of FIT in our cohort are practically identical to the findings of many already published diagnostic accuracy studies analyzing this biomarker. Therefore, our results using FIT are not providing new evidence.
The POC test, although has not shown a better overall diagnostic accuracy than FIT (this fact was already mentioned in the Discussion section), provides several advantages: less undetected cases (extensively analyzed in the Discussion section, probably the major drawback of FIT with the recommended 10 μgr/gr cut-off), ability to detect high-risk patients, and the specific advantages of a POC test (immediate results, easy to use). We believe that the validation of a POC test with these benefits and with a similar diagnostic accuracy than the most widely validated biomarker in this population (quantitative FIT) is the main finding of our study. However, the reviewer is right and the fact that the POC test has not a significant better diagnostic performance than FIT should be highlighted, so we have mentioned this in the Conclusions section.
Point 3. In Table 2, FC appears to have better sensitivity than FIT alone for CRC and adenomas - this would differ from other published series. What were the cut offs applied here?
More clarity and detail around this needs to be made in the Discussion
This is another interesting point. Evidence regarding FC diagnostic yield for CRC and adenomas is not as solid as in FIT, and sensitivity values vary widely between studies due to high heterogeneity in inclusion criteria, FC specific test used, and cut-offs. This scarce evidence has been summarized in a recent meta-analysis (Ross et al – ref 22), suggesting that FC may be a more sensitive but less specific biomarker for CRC diagnosis than FIT. However, the reviewer is right and in most comparative studies analyzing FC and FIT in symptomatic patients, FIT provided better sensitivity values for CRC diagnosis than FC (FIT 89.3% vs FC 82.1% in Mowat et al - ref 23 / FIT 87.5% vs FC 75% in Lué et al – ref 27). The cut-offs chosen in our study are 10 μgr/gr for FIT and 50 μgr/gr for FC, as it is stated in the Methods section. We have added a paragraph in the Discussion section mentioning this relevant information.
Point 4. Would be helpful to outline potential costs for POC and FIT as well as difference in uptake (%) by patients for faecal testing - this would determine if such testing can be applied more universally.
The reviewer correctly points out one of the main limitations of our study. We have not performed an economic analysis of these strategies based on faecal biomarkers. This evaluation would definitely add interest to our results, and we might perform it in future analyses of the dataset generated in this study. However, the main aim of this initial evaluation was to assess the diagnostic yield of these faecal markers in symptomatic population.
Although undoubtedly interesting, we cannot evaluate acceptation of faecal tests in general symptomatic population as all patients included in this study had previously accepted colonoscopy performance (which is usually worse accepted than faecal tests). We have added a paragraph in Discussion – Limitations section mentioning these points and adding two references bringing information about economic analyses performed in similar populations, and acceptance of faecal and endoscopic evaluations in symptomatic patients.
Point 5. Please add in the results section test accuracy against symptoms (rectal bleeding) – this was briefly discussed but not in detail
Also in the group with iron deficiency anaemia - this was a bigger cohort (32%)
As the reviewer correctly addresses, diagnostic accuracy of the faecal tests in specific symptoms is briefly mentioned in the Discussion (only for rectal bleeding as the indication of FIT in these patients may be criticized by readers), without presenting the whole diagnostic accuracy data in the Results section. We appreciate the reviewer noting this lack of information and we have added a paragraph in the Results section and two new Tables in the Supporting Information summarizing the main results of the faecal test stratified by symptoms.
Please note that 32.6% is the prevalence of significant pathology in patient with anemia (29/89) and not the relative frequency of this symptom in our cohort. The most frequent symptoms, as mentioned in the text (page 4), were previous history of rectal bleeding (28.9% - 165/571) followed by chronic diarrhea (20% - 114/571).

Round 2
Reviewer 2 Report
Thank you for providing detailed responses which I am satisfied with.
One minor point, as authors point out, study was not designed to look at drug interactions hence no need to go into detail around this especially as samples size is small.